# Activation and Regulation of Pancreatic Stellate Cells in Chronic Pancreatic Fibrosis: A Potential Therapeutic Approach for Chronic Pancreatitis

**DOI:** 10.3390/biomedicines12010108

**Published:** 2024-01-04

**Authors:** Fanyi Kong, Yingyu Pan, Dong Wu

**Affiliations:** 1Department of Gastroenterology, Peking Union Medical College Hospital, Chinese Academy of Medical Sciences and Peking Union Medical College, Beijing 100730, China; kongfy18@student.pumc.edu.cn (F.K.); panyy20@student.pumc.edu.cn (Y.P.); 2Clinical Epidemiology Unit, Peking Union Medical College Hospital, Chinese Academy of Medical Sciences and Peking Union Medical College, Beijing 100730, China

**Keywords:** chronic pancreatitis, pancreatic fibrosis, pancreatic stellate cells

## Abstract

In the complex progression of fibrosis in chronic pancreatitis, pancreatic stellate cells (PSCs) emerge as central figures. These cells, initially in a dormant state characterized by the storage of vitamin A lipid droplets within the chronic pancreatitis microenvironment, undergo a profound transformation into an activated state, typified by the secretion of an abundant extracellular matrix, including α-smooth muscle actin (α-SMA). This review delves into the myriad factors that trigger PSC activation within the context of chronic pancreatitis. These factors encompass alcohol, cigarette smoke, hyperglycemia, mechanical stress, acinar cell injury, and inflammatory cells, with a focus on elucidating their underlying mechanisms. Additionally, we explore the regulatory factors that play significant roles during PSC activation, such as TGF-β, CTGF, IL-10, PDGF, among others. The investigation into these regulatory factors and pathways involved in PSC activation holds promise in identifying potential therapeutic targets for ameliorating fibrosis in chronic pancreatitis. We provide a summary of recent research findings pertaining to the modulation of PSC activation, covering essential genes and innovative regulatory mediators designed to counteract PSC activation. We anticipate that this research will stimulate further insights into PSC activation and the mechanisms of pancreatic fibrosis, ultimately leading to the discovery of groundbreaking therapies targeting cellular and molecular responses within these processes.

## 1. Introduction

In the progression of fibrosis within the context of chronic pancreatitis (CP), a pivotal function is ascribed to pancreatic stellate cells (PSCs). These specialized cells, akin to fibroblasts, are positioned within the periacinar milieu of the pancreatic acini. Noteworthy attributes of PSCs include a central cellular body and extensive cytoplasmic extensions that intricately envelop the foundations of neighboring acinar cells [1].

In the context of a healthy pancreatic environment, PSCs exhibit a quiescent disposition, characterized by the presence of copious vitamin A-laden lipid droplets within their cytoplasm. These cells engage in a discerning pattern of gene expression, specifically highlighting desmin, glial fibrillar acidic protein (GFAP), vimentin, and nestin (intermediate filaments), alongside neuroectodermal markers like nerve growth factor (NGF) and neural cell adhesion molecule (NCAM), serving as distinctive biomarkers. Beyond this, PSCs demonstrate the capability to engage in the synthesis of diverse matrix metalloproteinases (MMP2, MMP9, and MMP13) as well as their regulatory counterparts TIMP1 and TIMP2. This dynamic orchestration of molecular processes contributes concertedly to the precise modulation of extracellular matrix (ECM) protein synthesis and degradation, thus upholding the integrity of the standard tissue architecture [1,2,3,4].

Nonetheless, within the distinctive microenvironment of CP, PSCs undergo a transformative process of activation, assuming a myofibroblast-like phenotype characterized by pronounced escalation in both cellular proliferation and migratory capability. Furthermore, this environment witnesses a dysregulation in the controlled release of metalloproteinases and their corresponding inhibitory factors. 

An expanding array of factors, operable through paracrine and autocrine pathways, is being progressively elucidated with regards to their stimulatory or inhibitory impact on PSC activation. Concomitantly, PSCs exhibit a proclivity for excreting an array of cytokines, chemokines, and growth factors. A subset of these factors registers notably elevated levels compared to their quiescent counterparts. Among this array are transforming growth factor-β (TGF-β), connective tissue growth factor (CTGF), monocyte chemoattractant protein-1 (MCP-1), interleukin (IL)-1, IL-6, IL-8, IL-15, IL-1β, IL-15, IL-33, and RANTES (regulated on activation normal T cell expressed and secreted) [5,6]. These inflammatory mediators significantly contribute to the self-triggered activation cascade of PSCs.

In the ultimate stages of this cascade, activated PSCs manifest a prolific output of α-smooth muscle actin (α-SMA). Additionally, these cells synthesize a substantial volume of ECM, which is predominantly localized around these cells. This repertoire encompasses key components such as type I collagen, type III collagen, fibronectin, and laminin [7]. Collectively, this orchestrated molecular machinery paves the way for the fibrotic process intrinsic to CP [8].

In the course of this complex process, a significant array of signal transduction pathways becomes intricately involved. Attempts to delve into the initiating signaling cascade responsible for the activation of PSCs, coupled with a comprehensive exploration of the ensuing cellular responses, stand as a pivotal stride toward the development of more precise therapeutic agents designed to impede pancreatic fibrosis. This investigative effort not only promises innovative approaches for addressing CP but also unveils promising avenues for the management of the associated disease. 

## 2. Activation and Contribution of PSCs to Fibrosis

Within the context of CP, PSCs respond to a multitude of stimulatory signals by undergoing activation, thus adopting the aforementioned activated state. Morphologically, these cells undergo a transformation akin to fibroblasts. At the molecular and cellular levels, a substantial secretion of immune factors and fibrotic precursors takes place, resulting in a significant contribution to pancreatic fibrosis. In this section, we delve into various aspects regarding PSC activation, encompassing physicochemical and pathogenic stimuli, the diverse array of cellular signals within the microenvironment of CP, as well as immune factor signals (illustrated in Figure 1). Furthermore, we elucidate the specific manifestations of PSCs following activation under distinct conditions, providing insights into their role in the progression of fibrosis in the context of CP. (The molecular and cellular characteristics of PSCs before and after activation is illustrated in Appendix A).

### 2.1. Alcohol-Induced Activation of PSCs

Roughly 70% to 90% of CP instances can be attributed to elevated alcohol consumption [9,10,11,12,13,14]. Alcohol-induced CP is recognized as a prominent risk factor, with a variety of elements orchestrating PSC activation throughout the continuum of chronic alcohol exposure. These elements include ethanol itself, along with its metabolite acetaldehyde [15,16], as well as fatty acid ethyl esters (FAEEs) [17,18], oxidative stress, and cytokines [19,20]. The enzymatic processing of alcohol by pancreatic acinar cells engenders oxidative stress, thereby furnishing a catalyst for PSC activation.

In the initial phases of alcoholic CP, there exists mild interstitial fibrosis within discrete acinar units, frequently accompanied by the emergence of PSCs around regions marked by tissue necrosis. As the condition advances into a state of multi-lobular fibrosis, the presence of PSCs is evident within the inter-lobular connective tissue. Throughout this evolving process, varying degrees of positive immunoreactivity are discernible for markers such as α-SMA, desmin, transforming growth factor-β receptor II (TGF-β-RII), and the platelet-derived growth factor receptor α isoform (PDGF-Rα). Conversely, the expression of the cytokine latency-associated peptide (LAP), the precursor of TGF-β, is diminished, while platelet-derived growth factor (PDGF)-B is notably absent [21].

Simultaneously, a substantial amount of evidence lends credence to the co-localization of 4-hydroxynonenal (4-HNE), the byproduct of lipid peroxidation, with activated PSCs. This phenomenon accentuates the intrinsic enzymatic capacity of PSCs to metabolize ethanol into acetaldehyde through their inherent ethanol dehydrogenase [19], thus instigating a cascade of oxidative stress and engendering their autonomous activation and lipid peroxidation [17]. This indicates that ethanol can activate PSCs from their quiescent state, rather than exclusively acting within the domain of activated PSCs [22]. This disclosure posits that during the early stages of chronic alcohol consumption, PSC activation is initiated and persists throughout the entire cycle of pancreatitis onset, including its pre-onset phase, continuously promoting pancreatic fibrosis. It is noteworthy that the antioxidant potential of vitamin E holds the capacity to forestall the activation of PSCs prompted by ethanol and acetaldehyde. Recent investigations also advance the notion that alcohol-induced injury to the pancreas could yield an amplified activation of the very low density lipoprotein receptor (VLDLR), consequentially inciting intracellular lipid accumulation and the ensuing development of dyslipidemia within PSCs. This intricate cascade, in turn, emerges as a pivotal driving force in the fibrotic progression associated with CP, concomitantly augmenting the expression and release of IL-33 within PSCs [23].

Alcohol and its metabolites induce the expression of α1(I)collagen, α-SMA, PDGF-Rβ, and TGF-β1 [17,24,25,26]. In addition to enhancing the formation of ECM, the above-mentioned products also elevate PDGF-induced NADPH oxidase activity, substantiating the theory of reactive oxygen species (ROS) playing a role in the activation of PSCs [22,27,28,29]. Due to activated PSCs being able to generate acetaldehyde and oxidative stress, activated PSCs also exhibit the expression of CCN2/CTGF, imparting them with the abilities of adhesion, migration, and collagen synthesis [27,29]. Evidence suggests that CCN2, conveyed through exosomes, can mediate paracrine collagen synthesis signaling to the surrounding PSCs [30].

The intracellular signaling mechanisms governing the activation of PSCs induced by ethanol have been meticulously elucidated. These mechanisms encompass pivotal pathways, including the mitogen-activated protein kinase (MAPK) pathway, phosphoinositide 3-kinase (PI3K), protein kinase C (PKC), and the transcription factor activator protein-1 (AP-1) [19,24,31]. Notably, ethanol acts in synergy to heighten the secretion of CX3CL1 from PSCs, a phenomenon attributed to the orchestrated activation of ERK and ADAM17. This chemokine, once self-secreted, effectively binds to the CX3CR1 receptor located on PSCs, thereby instigating their activation process [32,33].

Activated PSCs present a noteworthy mechanism involving cytoplasmic calcium ion overload. Ethanol, along with its non-oxidative metabolites, fatty acids, and FAEEs, influence quiescent PSCs by triggering the release of intracellular stored calcium ions. Consequently, Ca^2+^ from the extracellular fluid enters the cytoplasm, mostly through the CRAC/Orai1 channel, leading to intracellular calcium overload [20,34,35]. This process disrupts mitochondrial potential and triggers cell death [36]. To counteract excessive calcium overload, activated PSCs downregulate the transient receptor potential ankyrin 1 channel (TRPA1), conferring significant resistance to alcohol-induced cellular injury [37]. This explains the sustained fibrosis induction in the pancreas upon alcohol stimulation. Ethanol also augments lipopolysaccharide (LPS) endotoxin’s inhibitory effect on PSC apoptosis and promotes cell survival [38]. This, to some extent, maintains the population of activated PSCs, further propelling pancreatic fibrosis. In the early stages of CP, complete abstinence from alcohol can lead to the full reversal of pancreatic fibrosis. This effect likely arises from the removal of alcohol’s suppressive impact on PSC apoptosis, thereby halting the apoptosis of activated PSCs. This process encompasses fibrosis progression, encompassing ECM remodeling and immune dysregulation [21].

### 2.2. Cigarette Smoke-Induced Activation of PSCs

Smoking constitutes an independent risk factor to CP, and it can also act in conjunction with alcohol to expedite the fibrotic progression of this condition [39,40,41,42,43]. Clinically, this phenomenon is characterized by an escalation in both pancreatic calcification and fibrosis [44,45,46]. Tobacco use commonly triggers the inaugural episode of acute pancreatitis. Subsequent continuous exposure to smoking-induced injury can lead to recurrent pancreatitis [47,48]. Within this pathological course, activated PSCs initiate the process of pancreatic fibrogenesis, ultimately culminating in the establishment of CP in affected individuals [49].

Nicotine and its nicotine-derived nitrosamine, nicotine-derived nitrosamine ketone (NNK, formed during the tobacco drying process and also produced through nicotine metabolism in the body), induce the expression of the α7 subtype of the nicotinic acetylcholine receptor (α7nAChR) in cells. This phenomenon is not confined to pancreatic immune cells but has also been evidenced in PSCs. Ligands of the aryl hydrocarbon receptor found in cigarette smoke can upregulate the secretion of IL-22 from T cells within the microenvironment of CP in mice [50]. Consequently, IL-22 triggers the expression of ECM genes, including fibronectin 1 and collagen type I α1 chain, in PSCs [47].

Metabolites of NNK activate nuclear factor kappa-B (NF-κB), resulting in the release of tumor necrosis factor (TNF)-α from macrophages while inhibiting the synthesis of IL-10, thereby leading to the sustained activation of PSCs [51]. Meanwhile, exposure to nicotine significantly elevates intracellular and mitochondrial ROS levels in PSCs, thus promoting PSC activation by upregulating dynamin-related protein 1 (DRP1) and subsequently initiating mitochondrial fission [52].

Alcohol and cigarette smoke are capable of increasing the production of ROS within PSCs [18,53]. Components found in cigarette smoke, such as cigarette smoke extract (CSE) and the NNK, when combined with ethanol (EtOH) at clinically relevant concentrations, collectively activate PSCs. Notably, irrespective of the presence of an alcoholic milieu, both NNK and CSE significantly induce PSC activation through nAChRs. Apart from the consistent outcome of heightened PSC migratory capabilities, NNK exhibits an additional propensity to stimulate PSC proliferation and foster the secretion of type I collagen, an essential component of the ECM, whereas the projected effects of CSE and nicotine on type I collagen expression remain inconclusive [47]. Research suggests that the α7 isoform of nAChR also functions as a calcium channel. Upon nicotine stimulation, it could potentially lead to elevated intracellular calcium levels, either by permitting calcium influx from the extracellular environment into the cell [54] or by activating the α7 nAChR signaling pathway via G protein (Gαq) transduction, thereby activating the inositol trisphosphate receptor (IP3R) and subsequently releasing stored calcium from the local endoplasmic reticulum [55].

The intricate disruption of calcium ion equilibrium in PSC activation has been comprehensively discussed earlier. Meanwhile, nicotine also engages in PSC activation by triggering the α7nAChR-mediated JAK2/STAT3 signaling pathway. This, in turn, contributes to the increased synthesis of α-SMA and collagen, while concurrently suppressing the synthesis of TIMP1 and TIMP2, pivotal regulators of ECM remodeling [56].

### 2.3. Hyperglycemia-Induced Activation of PSCs

Research findings have consistently demonstrated that both hyperglycemia and hyperinsulinemia contribute cumulatively to the activation and proliferation of PSCs [57]. A study conducted by Ko et al. [58] highlighted the potential of high glucose to activate PSCs through the renin–angiotensin system. In this context, angiotensin II triggers DNA synthesis in PSCs via the transactivation of the EGF receptor and the activation of the ERK pathway [59]. Nomiyama et al.’s results indicated that high glucose might stimulate PSC activation through the PKC-p38 MAPK pathway [60]. Recent investigations have further shown that elevated glucose levels heighten oxidative stress, thereby facilitating PSC activation [61]. An analogue of glucagon-like peptide-1 (GLP-1) known as Exendin-4 (Ex-4) has been found to attenuate high-glucose-induced fibrosis by reducing angiotensin II and TGF-β1 production through the inhibition of ROS generation [62].

These studies collectively affirm the stimulating role of hyperglycemia in PSC activation and offer insights into the underlying mechanisms. Beyond its impact on PSC activation, hyperglycemia also induces the trans-differentiation of PSCs, fostering enhanced communication with cancer cells. This communication, in turn, activates MAPK signaling, subsequently promoting cancer cell proliferation [63].

Furthermore, it has been shown that glutathione can effectively curb oxidative-stress-induced PSC activation triggered by high blood glucose levels, both in vivo and in vitro. This intervention works by blocking the ROS/TGF-β/SMAD signaling pathway, thereby mitigating pancreatic fibrosis caused by oxidative-stress-induced PSC activation [64].

### 2.4. Pressure-Induced Activation of PSCs

Pancreatic fibrosis frequently accompanies prolonged obstruction of the pancreatic duct. Consequently, in CP, the tissue pressure within the pancreas exceeds that of a healthy pancreas. External pressure applied to the pancreas facilitates the activation and proliferation of PSCs, thereby boosting the production of MAPK proteins, α-SMA, and ECM components [65]. A study conducted by Asaumi et al. [66] indicated that pressure-induced activation could be attributed to the generation of intracellular ROS. More recent investigations have shifted their focus towards calcium ion channels, which respond to localized pressure and subsequently initiate downstream signaling events. In pressure-induced PSC activation, there is an observed increase in calcium influx mediated by the canonical transient receptor potential 1 (TRPC1) channel. TRPC1 has been identified as a regulator that sustains PSC activation via the ERK1/2 and SMAD2 pathways [67]. Another mechanosensitive calcium channel, Piezo1, is activated under high-pressure conditions and subsequently triggers the opening of transient receptor potential vanilloid-type 4 (TRPV4), thereby leading to PSC activation [68].

The sodium–calcium exchanger (NCX) plays a pivotal role in maintaining cellular Na^+^ and Ca^2+^ homeostasis. Studies have substantiated that NCX1 significantly contributes to the migratory behaviors of PSCs, with its impact intricately shaped by the specific cellular context in which it operates [69].

### 2.5. Acinar-Cell-Induced Activation of PSCs

Pancreatic fibrosis arises from the excessive accumulation of ECM components, such as collagen and fibronectin, within the pancreatic tissue. This phenomenon commonly emerges as a consequence of recurrent injuries experienced by individuals afflicted with chronic pancreatic ailments. The underlying pathophysiology is notably intricate, encompassing both the impairment of acinar cells and their subsequent stress responses. A thorough comprehension of the intricate role that acinar cells play in the context of pancreatic fibrosis holds paramount importance in the formulation of efficacious intervention strategies [70].

Within the microenvironment characterized by CP, compromised acinar cells exhibit considerable plasticity and heterogeneity [71]. These cells possess the capacity to directly activate PSCs, thereby acting as a cellular source driving fibrosis. Alternatively, they might indirectly contribute to the fibrotic process by liberating an array of substances or orchestrating the recruitment of immune cells, thereby fostering sustained activation of PSCs [72,73]. Furthermore, when acinar cells are co-cultivated with PSCs, a remarkable enhancement in migratory ability and the expression of ECM components in PSCs is observed [74].

In the context of previously mentioned alcoholic pancreatitis, the severe impairment of pancreatic exocrine acinar cells can trigger the activation of PSCs, leading to the initiation of intralobular fibrosis. This marks the initial phase of fibrotic development in CP [21]. The metabolic processing of ethanol by acinar cells generates ROS, a phenomenon that has been well documented. These ROS molecules have the potential to initiate the activation of the NF-κB and JAK/STAT signaling pathways within PSCs during the progression of acute pancreatitis, thereby fostering autocrine amplification [75]. Furthermore, injury to acinar cells due to exposure to both ethanol- and fatty-acid-conditioned medium stimulates the heightened expression of collagen and fibronectin in cultured [74] activated PSCs. Ethanol, in conjunction with the inhibitory effects arising from fatty acids produced through ethanol metabolism, can induce injury in acinar cells. This injury prompts the release of stored calcium ions within acinar cells, followed by the influx of activated extracellular Ca^2+^ into the cells. This cascade subsequently leads to an intracellular surplus of calcium ions [20,34,35], disrupting mitochondrial membrane potential and ATP generation [37], ultimately triggering cellular demise. Consequently, this sequence results in the generation of activated calcium signaling in PSCs, akin to the direct impact of alcohol on PSCs.

The mechanism of store-operated calcium entry (SOCE) mediated by Orai1 is a widely prevalent signaling pathway that, under pathological conditions, can become excessively activated, leading to an overload of intracellular calcium. A number of selective inhibitors of Orai1, such as CM5480 and CM4620, restore the expression of regulatory factors associated with SOCE in acinar cells, effectively mitigating uncontrolled calcium elevation [20]. This safeguarding mechanism not only preserves the functionality of acinar and ductal cells but also curtails immune cell infiltration and curbs the activation, proliferation, and migratory capabilities of PSCs [76].

When PSCs are treated with PIWI (P-element-induced wimpy testis) protein 1 (PIWIL1), facilitated by the PIWIL1-element, the consequential outcome is a substantial reduction in the expression levels of collagen I, collagen III, and α-SMA. Consequently, the invasive and migratory propensities of these cells are significantly curtailed. The PI3K/AKT/mTOR signaling pathway emerges as a plausible mechanism underlying the PIWIL1-mediated activation of PSCs. In light of this, PIWIL1 presents itself as a promising therapeutic target for addressing pancreatic fibrosis [77].

Acinar cells possess the capacity to produce CTGF at sites of injury [78,79]. CTGF assumes a significant role in activating PSCs by augmenting PSC proliferation and facilitating the secretion of chronic pro-inflammatory cytokines, including IL-1β [78,79]. Additionally, CTGF enhances the binding affinity of TGF-β1 to its receptor (both type Ⅰ and Ⅱ) by forming a complex with TGF-β1 [79,80]. This intricate interplay ultimately results in the direct or indirect induction of PSC collagen synthesis by CTGF [81]. Furthermore, a self-stimulatory feedback loop involving CTGF and TGF-β1 secretion is triggered within PSCs [80,82]. Research findings reveal that acinar cells’ Hippo pathway can impede the action of CTGF targeted by yes-associated protein 1 (YAP1) and transcriptional co-activator with PDZ-binding motif (TAZ). This inhibition effectively mitigates the aforementioned fibroinflammatory response without interfering with cell-autonomous proliferation [73,83]. Notably, acinar cells are capable of releasing cytokines such as TNF-α and TGF-β, thereby promoting their own fibrotic changes as well as those in PSCs [84,85]. Experimental evidence lends support to the notion that acinar cells express parathyroid hormone-related protein (PTHrP) during pancreatitis. Acting through paracrine pathways, PTHrP interacts with its receptor (PTH1R) on PSCs, eventually leading to the synthesis of ECM by PSCs and the secretion of pro-inflammatory factors. This dynamic process drives the fibrotic progression in CP [86,87]. The Wnt/β-catenin signaling pathway emerges as a pivotal participant in the course of pancreatic fibrosis and remodeling processes [88]. The Wnt signal is activated in damaged acinar cells, orchestrating acinar cell regeneration and proliferation, while beta-catenin activation of this pathway within both acinar cells and PSCs drives the fibrotic process [89,90]. Necrotic acinar cells release molecules associated with damage-associated molecular patterns (DAMPs), promoting the activation of PSCs. However, these observations are presently confined to experiments focused on acute pancreatitis [91]. Serving as carriers of endogenous microRNAs, exosomes assume a role in the pathological processes of diverse ailments. Exosome-borne miR-130a-3p sourced from acinar cells serves to activate PSCs and foster collagen formation by targeting peroxisome proliferator-activated receptor-gamma (PPAR-γ) within PSCs. Consequently, the suppression of miR-130a-3p offers a potential avenue for therapeutic intervention in the management of chronic pancreatic fibrosis [92].

### 2.6. Inflammatory-Cell-Induced Activation of PSCs

The development of pancreatic fibrosis is accompanied by the infiltration of inflammatory cells including lymphocytes, neutrophils, and macrophages [93]. Among them, macrophages have attracted the most attention. Macrophages are cells differentiated from monocytes, and commonly exist in two subsets, viz., classically activated or M1 macrophages, which are pro-inflammatory, and alternatively activated or M2 macrophages, which are anti-inflammatory [94,95]. There are shreds of evidence showing that macrophages are increased in CP tissues, among which M2 macrophages are dominant [96]. Macrophages in CP models and those from PSC cocultures express high levels of TGF-β and PDGF-β, suggesting that macrophages might participate in PSC activation directly through the paracrine release of cytokines and chemokines [96,97,98]. Moreover, macrophages in CP samples have a high expression of TIMP2 and MMP9, by which macrophages regulate ECM turnover [96]. A study shows that IL-6 produced by macrophages induces TGF-β1 production in PSCs, leading to PSC activation and collagen 1 synthesis [99]. On the other hand, PSCs isolated from mice and human patients with CP are found to express high levels of pro-inflammatory cytokines, IL-4 and IL-13, which promote alternative macrophage activation [100]. In response to activation of SARS-CoV-2 receptors, PSCs secrete IL-18, which acts on macrophages to generate calcium signals [101]. Therefore, not only do macrophages influence PSCs, PSCs can also activate macrophages. Ultimately, a positive feedback loop forms, which leads to pancreatic fibrosis and damage.

Other than macrophages, an increased number of lymphocytes have also been discovered in CP samples. CD8+ T cell- or NK cell-mediated cytotoxicity might play a role in the pathogenesis of CP [102]. In addition, mast cells, dendritic cells, monocytes, eosinophils, and B cells are also believed to be involved in the development of CP [93].

There is a potential connection between PSCs and IgG4-related diseases such as autoimmune pancreatitis. It is proposed that an unusual interaction occurs between immune regulators and PSCs, generating TGF-β, IL-10, and vitamin D, which stimulates the development of IgG4-producing plasma cells but suppresses other immune reactions. Furthermore, PSCs create a “tolerogenic” environment, which is characterized by cytokines like IL-10 and IL-21, as well as vitamins A and/or D. Regulatory immune cells, such as Tregs and Bregs, are attracted and entrapped, resulting in the differentiation of IgG4-switched B cells to plasma cells. The ongoing mutual activation between immune regulators and PSCs is suggested to contribute to the pathology of IgG4-related diseases [103].

### 2.7. Typical Regulatory Factors of PSC Activation

TGF-β emerges as a preeminent regulatory cytokine in orchestrating fibrotic responses. Heightened expression of TGF-β has been discerned in damaged acinar cells and platelets situated proximate to fibrotic areas [104,105]. Prevailing consensus supports the notion that TGF-β1 is a principal instigator of pancreatic fibrosis, acting through the activation of PSCs. In the early phases of CP, TGF-β1 stimulates PSCs, prompting an upsurge in the synthesis of α-SMA, the deposition of collagen types I and III, fibronectin, and laminin, thereby mediating the progression of pancreatic fibrosis [97,105,106,107]. Recent investigations unveil that TGF-β1-activated members of the MAPK family, JNK1 and ERK1, are markedly elevated within PSCs, triggering the heightened expression of α-SMA and fibronectin [108]. Activation of TGF-β receptors triggers the C-terminal phosphorylation of Smad2 and Smad3, which subsequently bind to cytoplasmic Smad4. The oligomeric Smad2-Smad4 and Smad3-Smad4 complexes are then translocated to the nucleus, stimulating the transcription of TGF-β target genes, and hence, promoting the activation of PSCs [109].

Meanwhile, TGF-α orchestrates an elevation in MMP-1 and MMP-2 levels within PSCs, thereby enhancing their migratory capacity through the degradation of collagen within the cellular basement membrane [110]. Furthermore, TGF-β exerts a safeguarding influence on the recently identified membrane-anchored MMP inhibitor, RECK, curbing its degradation. This mechanism effectively dampens MMP activity while concurrently fostering the deposition of ECM [111]. It is notable that PSCs themselves synthesize TGF-β1, indicating the existence of an autocrine loop that perpetuates PSC activation subsequent to the initiation of external signaling [98]. Both TGF-β1 and IL-1β, subjects to be explored subsequently, operate within a Smad3- and ERK-dependent autocrine loop [5]. In addition, TGF-β1 triggers the activation of the NF-κB pathway within PSCs by modulating the phosphorylation of TGF-β1-activated kinase 1 (TAK1). This mechanism propels NF-κB to foster the advancement of CP [112].

The receptor protein osteogenic protein-1 (OP-1) is a constituent of the TGF-β superfamily. It possesses the capacity to impede autocrine activin A activation and decrease TGF-β expression and secretion within PSCs, thereby repressing PSC activation and the release of collagen [113]. Treatment involving all-trans retinoic acid (ATRA) can obstruct the mechanical release of active TGF-β by PSCs, attenuate TGF-β’s bioactivity, and consequently impede the myofibroblast phenotype in active PSCs [107]. Bone morphogenetic proteins (BMPs), which are also members of the TGF-β superfamily, possess the capability to counteract TGF-β’s fibrotic function in PSCs via the Smad1/5 signaling pathway. Pre-treatment with BMP2 inhibits TGF-β-induced expression of α-SMA, fibronectin, and collagen Ia, indicating its ability to mitigate TGF-β’s fibrogenic impact on PSCs [114]. Hydrogen peroxide-inducible clone-5 (Hic-5) acts as an activator of hepatic stellate cells (HSCs), and during the transition of PSCs from the quiescent to myofibroblast-like phenotype, it governs TGF-β’s pro-promoting influence on PSCs by stimulating Smad2 phosphorylation. A deficiency in Hic-5 may present a potential therapeutic target for fibrosis in cases of CP [115].

Both activin A and TGF-β are autocrine activating factors for PSCs. Activin (A) receptors I (ActRI, also termed ALK4) and activin receptor type IIa (ActRIIa) are present on PSCs, and their reciprocal enhancement of expression has been demonstrated. Furthermore, activin A, in a dose-dependent manner, collaborates with TGF-β1 to amplify PSC activation and collagen secretion [113]. Angiotensin II (Ang II) facilitates the proliferation of activated PSCs by activating epidermal growth factor receptors. This phenomenon involves inducing the expression of Smad7 through a PKC-dependent pathway, consequently inhibiting autocrine TGF-β1-mediated suppression of PSC growth and augmenting the proliferation of activated PSCs [116]. Fibroblast growth factors FGF-1 and FGF-2 were observed to be upregulated during the later phases of the experimental model for CP, stimulating the expression of fibronectin and type I collagen mRNA within PSCs [106]. Additionally, basic fibroblast growth factor (bFGF) possesses the capacity to induce PSC proliferation in vitro [117].

CTGF is a member of the CCN family of proteins. Upon binding to the α5β1 integrin, CTGF enhances the activation of PSCs [79]. It can regulate CP fibrosis through acinar cell secretion or PSC autocrine mechanisms [78]. Its expression in PSCs can be augmented by TGF-β1, activin A, and TNF-α [1,81,117]. Within PSCs, CTGF targets genes involved in ECM protein synthesis, as well as the genes of pro-inflammatory cytokines IL-1β and IL-6. Moreover, CTGF promotes PSC proliferation, thereby collectively accelerating the progression of chronic inflammation through the NF-κB pathway. Similar to activin A, CTGF can be rapidly downstream-stimulated by high levels of TGF-β1 in PSCs [82,118]. Additionally, CTGF contains a cysteine-rich domain (CR) that interacts with the corresponding domains of BMP and TGF-β. This interaction hinders BMP4 from binding to its receptor while enhancing the binding of TGF-β1 to its receptor [80].

Interleukins are a class of immunoregulatory mediators that play vital roles in inflammatory responses. Currently, their effects on PSC activation manifest with a diverse spectrum of promotive or inhibitory actions. IL-10 is a potent anti-inflammatory and anti-fibrotic factor that demonstrates strong efficacy in liver inflammation and acute pancreatitis [119]. Studies have shown that IL-10 can limit plasma TGF-β1 levels and activated PSC numbers during recurrent acute pancreatitis episodes, effectively restraining the production of type I and type III collagen proteins and mitigating potential fibrosis and glandular atrophy processes associated with CP [120]. IL-10 has no impact on PSC proliferation, although reports suggest its potential to promote collagen and α-SMA synthesis [121]. Activated PSCs express IL-33 in their nuclei, a process promoted by IL-1β through pathways including NF-κB, ERK, p38 MAPK, and PDGF-BB via the ERK pathway. IL-33 may regulate PDGF-induced PSC proliferation [122]. IL-13 inhibits PSC proliferation by suppressing NF-κB transcriptional activity, leading to reduced autocrine TGF-β1 production [123]. Other cytokines, such as PSC autocrine factor IL-1, promote α-SMA synthesis without affecting collagen production. IL-6 promotes α-SMA synthesis while inhibiting PSC proliferation and collagen deposition [6]. However, it is noteworthy that IL-6, via the transmembrane receptor gp130, activates the Jak/STAT pathway, particularly Jak2/STAT3, to govern genes associated with survival and inflammation. Inhibition of Jak1/2, such as with ruxolitinib, reduces STAT3 phosphorylation and cell proliferation, thus suppressing this process [86]. TNF-α promotes collagen deposition, α-SMA synthesis, and the proliferation of PSCs. Activators of PSCs also include IL-8, MCP-1, and RANTES, where IL-8 and MCP-1 secretion is upregulated by IL-1β and TNF-α, and RANTES secretion is mainly induced by TNF-α [6]. The origins and actions of these inflammatory factors are multifaceted, originating from immune cells, acinar cells, and even PSCs themselves. The use of PDTC and TPCK to inhibit NF-kB activation has significantly reduced the mRNA expression of chemotactic factors induced by IL-1β and TNF-α. PDTC and TPCK are potent inhibitors of NF-kB activation induced by IL-1β or TNF-α [6].

PDGF secreted by platelets has been reported as a proliferative factor secreted by platelets for PSCs [28,97,105]. It can induce rapid activation of Raf-1, ERK 1, and ERK 2, as well as AP-1 protein, thereby promoting mitosis in PSCs [109]. PDGF leads to ECM synthesis in PSCs and, among them, PDGF-BB is the most potent mitogen for PSCs. Activated PSCs express PDGF α and β receptors, and PDGF-BB induces autophosphorylation of its receptor, subsequently activating the PI3-K, Akt, and ERK pathways, endowing PSCs with high migration and proliferation capacities [124]. PDGF receptors on PSCs can also be upregulated by TGF-β, wherein PSCs increase proliferation and collagen synthesis in response to the cytokines PDGF and TGF-β [97]. Moreover, PDGF can stimulate the activation and migration of PSCs through the TRPC3 and KCa3.1 channels, characterized by Ca^2+^ influx in PSCs [125].

Endothelin-1 and the pro-inflammatory chemokine CX3CL1, similar to PDGF, are also activators of PSCs. PSCs express ET(A) and ET(B) receptors, and upon ET-1 stimulation, cytoplasmic Ca^2+^ levels increase, enhancing PSC contraction and migration without significant proliferation [126,127]. External stimuli can induce PSC activation and CX3CL1 release [32], promoting the proliferation of activated PSCs through its receptor, CX3CR1, rather than secretion of inflammatory factors [33]. Furthermore, it is noteworthy that human PSCs conspicuously express cyclooxygenase-2 (COX-2) and proficiently synthesize prostaglandin E2 (PGE2). Notably, PGE2 plays a pivotal role in orchestrating several critical processes within PSCs, encompassing cellular proliferation, migration, and invasion, as well as the upregulation of genes associated with ECM and MMPs. This intricate cascade of events is intricately modulated through the mediation of EP4 receptors resident in PSCs, a relationship substantiated and validated through rigorous experimentation employing targeted receptor antagonists [128]. The stimulatory effect of fibrosis itself on PSCs is often overlooked. Fibrinogen directly stimulates the production of IL-6, IL-8, MCP-1, vascular endothelial growth factor, angiopoietin-1, and type I collagen in PSCs. It increases the expression of α-SMA and activates NF-κB, Akt, and three categories of mitogen-activated protein kinases [129]. Wnt2 and β-catenin are also significantly increased factors in pancreatic fibrosis, which are highly expressed in PSCs. They have been shown to elevate the expression of collagen 1α1, TGFβRII, PDGFRβ, and α-SMA in PSCs. Meanwhile, Dickkopf-related protein 1 (Dkk1) upregulation induces apoptosis in PSCs [130]. Retinoic acid (RA), a vitamin A derivative, has diverse biological functions, including regulating cell differentiation and proliferation, and mitigating progressive fibrosis in various organs. RA also induces apoptosis in PSCs and reduces the extent of pancreatic fibrosis by downregulating Wnt2 and β-catenin [90]. Hence, Wnt2 and Dkk-1 might serve as potential therapeutic targets for CP.

## 3. Prospects for PSC Therapy

The various factors and pathways involved in PSC activation have been reviewed in the previous sections. Efforts have also been made to identify potential targets within these mechanisms to provide therapeutic approaches for the management of fibrosis in CP. Several factors with inhibitory effects on PSC activation have been discussed earlier [6,90,107,113,114,115,130]. Apte et al. have summarized the significant effects of drugs targeting oxidative stress, TGF-β inhibition, TNF-α inhibition, anti-inflammatory agents, and PSC activation signaling molecules on reversing pancreatic fibrosis in experimental models, which demonstrate the potential reversibility of early-stage pancreatic fibrosis [18]. In recent years, new studies have focused more on regulating PSC activation pathways, key genes involved in PSC activation, and on the use of novel regulatory mediators to counteract PSC activation, which offers broader relief for pancreatic fibrosis (summarized in Table 1).

Regarding the regulation of PSC activation pathways by novel regulatory mediators, FTY720 can attenuate chronic pancreatic fibrosis by inhibiting T-cell infiltration. It has been found that FTY720 suppresses PSC activation by promoting apoptosis and inhibiting autophagy. This is achieved by inhibiting AMPK and activating the mTOR pathway, which leads to a significant negative regulation of PSC survival, proliferation, and migration [131]. Docosahexaenoic acid (DHA) has anti-inflammatory properties, and recent research indicates that DHA inhibits cytokine expression and NF-κB activation induced by poly (I:C) or TNF-α by reducing intracellular and mitochondrial ROS levels in PSCs. Consumption of DHA-rich foods might help prevent CP by inhibiting cytokine expression in PSCs [132]. Inhibitory Smad proteins (I-Smads) inhibit TGF-β intracellular signaling in PSCs. Promoting I-Smads during CP creates a negative feedback loop that inhibits PSC activation, similar to the action of TGF-β receptor I kinase inhibitor SB431542 [133]. Activation of the NLRP3 inflammasome plays a significant role in the development of pancreatic fibrosis. During CP, NLRP3 is directly involved in PSC activation. The NLRP3 inhibitor MCC950 suppresses NLRP3-mediated PSC activation by inhibiting the TGF-β1/Smad3 pathway [134]. Hsp90 inhibitor 17AAG can degrade TGFβRII within PSCs through the ubiquitin-proteasome pathway, thereby blocking the TGFβR-mediated Smad2/3 and PI3K/Akt/GSK-3β signaling pathways. This inhibits TGFβ1-induced PSC activation and ECM accumulation, making it a potential therapeutic strategy for pancreatic fibrosis [135]. Puerarin, a flavonoid derived from the traditional Chinese medicine kudzu root, exhibits anti-fibrotic effects in various fibrotic diseases. Experimental evidence indicates that puerarin significantly inhibits phosphorylation of MAPK family proteins (JNK1/2, ERK1/2, and p38 MAPK) in PSCs in a dose-dependent manner, making it a promising candidate for targeting the MAPK pathway [136]. In CP, the impaired retinoic acid receptor-related orphan receptor A (Rora)/nuclear receptor subfamily 1, group D, member 1 (Nr1d1)/aryl hydrocarbon receptor nuclear translocator-like (Arntl or Bmal1) loop, known as the circadian stabilizing loop, contributes to the fibrogenic properties of PSCs. Disruption of the balanced antagonistic action between Nr1d1 and Rora due to PSC activation leads to cytoplasmic loss of retinol-laden lipid droplets. The administration of melatonin and Rora agonist SR1078 can restore the stability of the circadian stabilizing loop, thereby stabilizing PSC regulation of extracellular mechanisms [137]. The antioxidant Mitoquinone (MitoQ) balances free radical levels and intracellular antioxidant systems in activated PSCs, inhibiting PSC activation and subsequent fibrotic phenotype development [138]. This finding aligns with the description by Apte et al. [18]. The collagen family receptor tyrosine kinase receptors DDR1/DDR2 inhibitor Imatinib (IMT) suppresses ECM deposition and PSC activation in CP. IMT also inhibits the TGF-β1/Smad signaling pathway, making it a promising candidate for CP [139]. Sphingosine-1-phosphate (S1P) is a biologically active lipid molecule that regulates various functions through its receptors (S1PR) present in different cells. The binding of S1P to S1PR2 promotes PSC activation and pancreatic fibrosis by regulating autophagy and the NLRP3 inflammasome. These findings provide a theoretical basis for targeting the S1P/S1PR2 axis in the treatment of pancreatic fibrosis [140]. Toll-like receptors (TLRs) regulate the transition of stellate cells. Excessive expression of TLR5 is required for TGF-β-mediated activation of PSCs. Regulation of TLRs could alleviate the effects of TGF-β [141]. Nintedanib (Ninte), approved for treating pulmonary fibrosis, also inhibits PSC activation and proliferation through the JAK/STAT3 and ERK1/2 pathways. These findings suggest that Ninte may constitute a potential anti-inflammatory and anti-fibrotic treatment for CP [142].

In terms of the key genes involved in PSC activation, secreted protein acidic and rich in cysteine (SPARC) is an extracellular glycoprotein involved in tissue remodeling, embryonic development, and tumor progression. It binds to various ECM components and is highly expressed in PSCs during the recovery phase of recurrent acute pancreatitis and CP, along with a high expression of fibrotic phenotype markers (collagen Ⅰ, fibronectin). SPARC is not only a potential therapeutic target but also a potential biomarker for the progression of pancreatic diseases [143]. Long non-coding RNA small nucleolar RNA host gene 11 (SNHG11) is highly expressed in the plasma of CP patients and TGF-β1-treated PSCs. SNHG11 regulates leukemia inhibitory factor (LIF) expression by sequestering miR-34b. Considering the promotive role of miR-34b in PSC proliferation, migration, and matrix accumulation induced by TGF-β1, silencing SNHG11 through the miR-34b/LIF axis, shows promise for treating CP [144]. Long non-coding RNAs (LncRNAs) have also been recognized as key regulators of fibrosis-related diseases. One such molecule, Lnc-PFAR, upregulates RB1CC1-induced autophagy by reducing miR-141 expression, thereby promoting PSC activation and pancreatic fibrosis [145]. In addition, miR-301a, a pro-inflammatory microRNA activated by various inflammatory factors in the tumor microenvironment, maintains PSC activation and fibrosis through the Tsc1/mTOR and Gadd45g/Stat3 pathways. Silencing miR-301a has the potential to alleviate fibrosis [146]. Replacing miR-15a with 5-FU-miR-15a through miRNA modification significantly reduces PSC viability, proliferation, and migration, concurrently downregulating YAP1 and BCL-2 levels. This results in promising targeting of CP using ectopic transfer of miRNA mimics [147]. Reg proteins stimulate PSC activation. Global deletion of the Reg1–3 (Reg1, 2, 3a, 3b, 3d, 3g) genes in mice leads to reduced pancreatic parenchyma loss, decreased collagen deposition, and a lower expression of inflammatory cytokines in CP. Regulation of human Reg gene expression could become a key element in future CP therapy [148]. The transcription factor JUN is crucial for maintaining the quiescent state of PSCs, providing a theoretical foundation for the treatment of various pancreatic injuries caused by PSCs [149].

## 4. Conclusions

CP is widely regarded within the medical field as a progressively debilitating condition characterized by sustained and irreversible damage to the pancreas. This damage encompasses multifaceted impairments such as acinar cell necrosis and the deposition of fibrotic substances within the pancreatic interstitium. Emerging research emphasizes the pivotal role of PSCs in orchestrating this process.

Our review aims to comprehensively conclude the post-activation behavior of CP, particularly delineating its influence on modulating the balance of ECM deposition. A succinct overview of the activation of PSCs is instrumental in unraveling the etiology and pathogenesis of CP. We expound upon the fibrotic mechanisms of PSCs in this context, considering both micro- and macro-factors, including alcohol consumption, smoking, hyperglycemia, mechanical stress, acinar cell injury, and the role of inflammatory cells.

Our focus extends to the regulatory factors and pathways governing the activation of PSCs, encompassing pivotal cellular factors such as TGF-β, CTGF, interleukins, and PDGF, alongside their involvement in PSC activation pathways. The conclusion of this article delves into targeted drugs aimed at mitigating these regulatory factors and genes, thereby providing foundational medical evidence for the development of therapeutic interventions inhibiting PSC activation in CP.

The integration of prevalent immunotherapy and cell-based therapies, such as pre-clinically studied anti-inflammatory effects of mesenchymal stem cells and Treg cells for pancreatic protection and repair in damaged tissues, suggests that intervening in PSCs holds promise in augmenting clinical management and improving prognostic outcomes for CP patients.

Pancreatic ductal adenocarcinoma (PDAC) is a lethal disease with poor prognosis, which accounts for over 90% of all pancreatic tumors. Multiple studies have shown that CP is an established risk factor for PDAC, both of which involve PSC activation and pancreatic fibrosis [150,151,152]. Therefore, preventing or curing CP is also an effective measure to reduce the incidence of PDAC, and limiting the activation of PSCs in the early stages of CP holds the prospect to improve the carcinogenic potential in patients with early-stage CP.

## Figures and Tables

**Figure 1 biomedicines-12-00108-f001:**
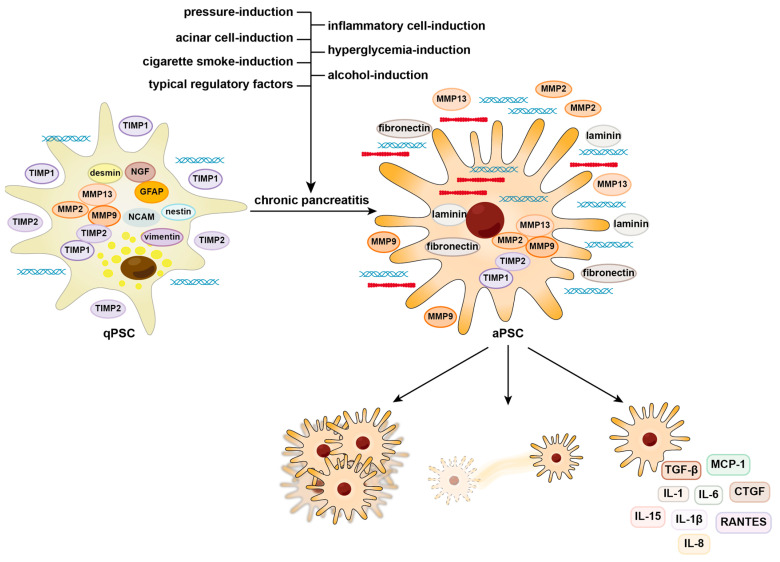
This figure illustrates the molecular and cellular characteristics of pancreatic stellate cells before and after activation. Activated pancreatic stellate cells exhibit functions including migration, proliferation, and significant secretion of autocrine activating factors. Some of the activation triggers for pancreatic stellate cells are also summarized and categorized. Red and blue coarse and fine helical shape: extracellular matrix components; yellow circular shape: vitamin A lipid droplets; elliptical shape: pancreatic cell-state marker; rounded square shape: autocrine activating factors.

**Table 1 biomedicines-12-00108-t001:** Summary of regulatory agents counteracting PSC activation and their mechanisms.

Regulatory Mediators	Primary Mechanisms of Agents on Pancreatic Stellate Cells
FTY720	FTY720 activates the mTOR pathway to mediate apoptosis in PSCs.
DHA	DHA reduces ROS in PSCs to inhibit the expression of autocrine cytokines.
I-Smads	I-Smads inhibit intracellular signaling of TGF-β in PSCs, and the activation of Smad6 and Smad7 can downregulate activation markers in PSCs.
MCC950	MCC950 inhibits the activation of NLRP3 in PSCs by suppressing the TGF-β1/Smad3 pathway.
17AAG	17AAG degrades TGFβRII through the ubiquitin-proteasome pathway, thereby inhibiting TGFβ1-induced activation of PSCs and extracellular matrix accumulation.
Puerarin	Puerarin inhibits the activation of MAPK family proteins (JNK1/2, ERK1/2, and p38 MAPK) in PSCs.
Mitoquinone (MitoQ)	MitoQ balances free radical levels and intracellular antioxidant system levels, thereby inhibiting PSC activation.
Imatinib (IMT)	IMT inhibits the TGF-β1/Smad signaling pathway in PSC activation.
S1P	Modulating the binding of S1P to S1PR2 can regulate autophagy and NLRP3 inflammasome-promoted activation of PSCs.
TLRs	Regulation of Toll-like receptors (TLRs) can mitigate the effects of TGF-β.
Nintedanib (Ninte)	Ninte inhibits the activation and proliferation of PSCs through the JAK/STAT3 and ERK1/2 pathways.
SPARC	In the recovery phase of recurrent acute pancreatitis and during the activation of PSCs in chronic pancreatitis, the SPARC gene is highly expressed.
miR-34b	The silencing of SNHG11 attenuates PSC proliferation, migration, and matrix accumulation through the miR-34b/LIF axis.
miR-141	Regulation of long non-coding RNAs (LncRNAs) may play a role in controlling PSC autophagy and activation.
miR-301a	The silencing of miR-301a mediates effective inhibition of the Tsc1/mTOR and Gadd45g/Stat3 pathways, thereby maintaining PSC activation and fibrosis.
miR-15a	The miRNA modification of miR-15a to 5-FU-miR-15a significantly reduces the viability, proliferation, and migration of PSCs.
Reg1-3	Deletion of the Reg gene in mice can lead to reduced pancreatic parenchyma loss, decreased collagen deposition, and reduced expression of inflammatory cytokines in chronic pancreatitis.
JUN	JUN is a key transcription factor in maintaining the quiescent state of PSCs.

## Data Availability

The data underlying this article are available in the article.

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
