# Peer review of "Activation and Regulation of Pancreatic Stellate Cells in Chronic Pancreatic Fibrosis: A Potential Therapeutic Approach for Chronic Pancreatitis"

_biomedicines, 2024, doi:10.3390/biomedicines12010108_

Round 1

Reviewer 1 Report

Comments and Suggestions for Authors

This is generally a well written article that helpfully summarizes a lot of recent information. A few schematic diagrams would, however, be helpful and further improve readability (Fig. 1 is completely unreadable in the ms version I have been able to download).

There are a number of specific issues that need to be addressed in a revised version:

[1]         In line 101, reference 33 should also be cited.

[2]         Line 152: There is no such thing as ‘activated Ca2+’! Delete ‘activated’.

[3]         Lines 152/153: It would be helpful here to mention that Ca2+ influx mostly occurs through CRAC/Orai1 channels (Reference 33).

[4] In the paragraph lines 285-292, it should be mentioned that a number of small molecule inhibitors of CRAC/Orai1 channels  have been shown to inhibit Ca2+ entry (reference 33). CM5480 is just one of them.

[5] In the paragraph lines 332-348, it should be mentioned that in addition to macrophages influencing stellate cells, there is also evidence showing that stellate cells can activate macrophages (Gerasimenko et al Function 2022).

Author Response

Respected Reviewer,

I sincerely appreciate your valuable editorial suggestions. I have duly acknowledged and incorporated all your inquiries, making the requisite modifications to the manuscript as per your guidance. The revised sections have been appropriately highlighted for your convenience.

Specifically, I have addressed your concerns on multiple fronts: I have marked all references to the original citation 33, with particular emphasis on line 101; emphasized near line 156 that Ca2+ influx predominantly occurs through CRAC/Orai1 channels; underscored at line 292 the significance of small molecule inhibitors of CRAC/Orai1 channels, such as CM5480 and CM4620, in restoring the expression of regulatory factors associated with SOCE in acinar cells; and, at line 352, elucidated the response of pancreatic stellate cells (PSCs) to the activation of SARS-CoV-2 receptors, wherein PSCs secrete IL-18, leading to the generation of calcium signals in macrophages. Thus, the mutual influence between macrophages and PSCs, including the activation of macrophages by PSCs, has been duly clarified.

Once again, I express my gratitude for your insightful feedback and the time you have dedicated to reviewing my manuscript. Your guidance is highly valued, and I welcome any further suggestions or comments at your convenience.

Best regards,

Fanyi Kong

Reviewer 2 Report

Comments and Suggestions for Authors

This is solid work that gives an exhaustive overview over the role of pancreatic stellate cells in chronic pancreatic fibrosis.

Lines 22 and 590: „TGF“ needs to be specified. TGF-β or TGF-a?

Line 64: „localized“ requires the addition of a specific site.

Lines 68/69 and 467/468: The same sentence contains the term „intricate“ twice. This sounds clumpsy.

Line 82: Please change „stimulus“ to „stimuli“ and remove „signals“.

Line 83: The fullstop needs to be placed behind the bracket. Please replace „Picture“ by „Figure“, „scheme“ or „cartoon“.

Line 151: Remove the „s“ from „influences“.

Lines 265-7: add „in PSCs“ after „components“.

Line 304: To which TGF-beta receptor do the authors refer here? Type I or II?

Line 349: Please add „numbers of“ between „increased“ and „lymphocytes“.

Line 361: Please replace „mediated“ by either „activated“ or „stimulated“.

Lines 363-5. Please rephrase this sentence. The content is wrong! „Activation of TGF-β receptors triggers the C-terminal phosphorylation (= activation) of Smad2 and Smad3, which subsequently bind to cytoplasmic protein Smad4. The oligomeric Smad2-Smad4 and Smad3-Smad4 complexes are then translocated to the nucleus, where they stimulate the transcription of TGF-β target genes.

Line 385: Replace „Smad1“ by „Smad1/5“.

Line 393: The term „Activin A receptors“ is misleading as the receptors bind all three activin isoforms, activin A, AB, and B. This term should be replaced by activin (A) receptor type I (ActRI, also termed ALK4) and activin receptor type IIa (ActRIIa).

Line 394: Please replace „within“ by „on“.

Line 446: Please move „secreted by platelets“ to behind „PDGF“.

Line 447: Please remove „s“ from „ERKs“.

Lines 475,476,478: The term „upregulate“ appears three times in a row. This sounds clumpsy.

Line 490: Please remove „at“.

Line 507: The sentence „Inhibiting inhibitory Smad proteins (I-Smads) disrupts TGF-β intracellular signaling in PSCs.“ is confusing as I-Smads normally inhibit TGF-β signaling. Please rephrase!

Table 1 needs a headline. The lettering in my copy is so diffuse and small that it is hardly readable. Please replace.

General

Abbreviations should be introduced at first use and subsequently used consequently and consistently. The inconsistent use of abbreviations is really a mess in this manuscript! This relates to the following terms (not limited to): PSCs, BMP, PDGF, CTGF, TNF-a, NFκB, MAPK, ROS, SMA, ECM, TGF-β, IL. The authors should activate the „search“ function in MS Word and correct this accordingly. They should also consider to abbreviate „chronic pancreatitis“ by „CP“. My estimate is that this will condense the length of the manuscript by 0.5 to 1 page.

Given the fact that chronic pancreatitis is an established risk factor for pancreatic ductal adenocarcinoma (PDAC), the authors should discuss (or at least mention) that preventing or curing CP is likely to also reduce the incidence of PDAC, thus representing a good strategy for cancer prevention.

Comments on the Quality of English Language

Moderate editing of English language required

Author Response

Respected Reviewer,

I deeply appreciate your invaluable editorial suggestions. I have duly received and implemented all your queries, incorporating the recommended modifications into the manuscript, which have been appropriately highlighted. The detailed expressions throughout the paper have been meticulously revised according to your guidance.

In the content of the article, I have added a statement at the conclusion, highlighting that pancreatic ductal adenocarcinoma (PDAC) is a lethal disease with poor prognosis, constituting over 90% of all pancreatic tumors. Chronic pancreatitis (CP) is a established risk factor for PDAC, both involving the activation of pancreatic stellate cells (PSCs) and pancreatic fibrosis. The inhibition of PSC activation, leading to chronic pancreatitis, is closely associated with the prevention of PDAC. Given that the primary focus of this article is on exploring the role of PSC activation in CP, extensive details on PDAC were not included. Your suggestions have been instrumental in guiding my discussion on limiting PSC activation to decrease the carcinogenic potential in patients with early-stage CP.

Once again, I express my gratitude for your constructive feedback and the time you dedicated to reviewing my manuscript. Your insightful input has enhanced the rigor of my manuscript. Please feel free to provide any further guidance at your convenience.

Best regards,
Fanyi Kong

Reviewer 3 Report

Comments and Suggestions for Authors

Authors prezented exhaustive review on the potential mechanisms involving stellate Cells in the chronic pancreatitis. The article is well-executed, but I would like to ask authors to provide some details on potential connections between stellate cells in IgG4 disease and their participation in genetic forms of pancreatitis (e.g. PRSS mutations)

Minor issue detected:

Figures in the PDF are of low resolution/compressed this should be addressed in the final version of the manuscript. 

Author Response

Respected Reviewer,

I sincerely appreciate your valuable editorial suggestions. I have duly received and incorporated all your queries into the manuscript, with the respective modifications being highlighted. In the revised manuscript, we have provided additional information on the role of pancreatic stellate cells (PSCs) in IgG4 pancreatitis, as outlined in line 361. Regarding the potential connections between stellate cells in IgG4 disease and their participation in genetic forms of pancreatitis, particularly PRSS mutations, we have learned that PRSS mutations are primarily associated with hepatic fibrosis and the functioning of hepatic stellate cells (HSCs), and they are not commonly observed in chronic pancreatitis (CP). Therefore, we regret to inform you that details on PRSS mutations were not included in the manuscript. We appreciate your understanding on this matter.

Once again, I express my gratitude for your insightful feedback and the time you dedicated to reviewing my manuscript. Your suggestions have contributed to the enhanced precision of my manuscript. Please feel free to provide any further guidance at your convenience.

Best regards, 

Fanyi Kong